# SETD2 suppresses tumorigenesis in a KRAS[G12C]-driven lung cancer model, and its catalytic activity is regulated by histone acetylation

Ricardo J Mack[1,2†], Natasha M Flores[3†], Geoffrey C Fox[4], Hanyang Dong[1], Metehan Cebeci[1], Simone Hausmann[3], Tourkian Chasan[3], Jill M Dowen[4,5,6,7,8], Brian D Strahl[4,5,8], Pawel K Mazur[3*], Or Gozani[1,2*]

[1]Department of Biology, Stanford University, Stanford, United States; [2]Cancer Biology Training Program, Stanford University, School of Medicine, Stanford, United States; [3]Department of Experimental Radiation Oncology, The University of Texas MD Anderson Cancer Center, Houston, United States; [4]Curriculum in Genetics and Molecular Biology, University of North Carolina at Chapel Hill, Chapel Hill, United States; [5]Department of Biochemistry and Biophysics, University of North Carolina at Chapel Hill, Chapel Hill, United States; [6]Department of Biology, University of North Carolina at Chapel Hill, Chapel Hill, United States; [7]Integrative Program for Biological and Genome Sciences, University of North Carolina at Chapel Hill, Chapel Hill, United States; [8]Lineberger Comprehensive Cancer Center, University of North Carolina at Chapel Hill, Chapel Hill, United States

*For correspondence:
pkmazur@mdanderson.org (PKM);
ogozani@stanford.edu (OG)

†These authors contributed equally to this work

## eLife Assessment

This is a **fundamental** study providing molecular insight into how cross-talk between histone modifications regulates the histone H3K36 methyltransferase SETD2. The article contains excellent quality data, and the conclusions are **convincing** and justified. This work will be of interest to many biochemists working in the field of chromatin biology and epigenetics.

**Abstract** Histone H3 trimethylation at lysine 36 (H3K36me3) is a key chromatin modification that regulates fundamental physiological and pathological processes. In humans, SETD2 is the only known enzyme that catalyzes H3K36me3 in somatic cells and is implicated in tumor suppression across multiple cancer types. While there is considerable crosstalk between the SETD2-H3K36me3 axis and other epigenetic modifications, much remains to be understood. Here, we show that Setd2 functions as a potent tumor suppressor in a KRAS[G12C]-driven lung adenocarcinoma (LUAD) mouse model, and that acetylation enhances SETD2 in vitro methylation of H3K36 on nucleosome substrates. In vivo, Setd2 ablation accelerates lethality in an autochthonous KRAS[G12C]-driven LUAD mouse tumor model. Biochemical analyses reveal that polyacetylation of histone tails in a nucleosome context promotes H3K36 methylation by SETD2. In addition, monoacetylation exerts position-specific effects to stimulate SETD2 methylation activity. In contrast, mono-ubiquitination at various histone sites, including at H2AK119 and H2BK120, does not affect SETD2 methylation of nucleosomes. Together, these findings provide insight into how SETD2 integrates histone modification signals to regulate H3K36 methylation and highlights the potential role of SETD2-associated epigenetic crosstalk in cancer pathogenesis.

## Introduction

Protein lysine methylation is a common post-translational modification (PTM) that occurs in three distinct states—monomethylation (Kme1), dimethylation (Kme2), and trimethylation (Kme3)—depending on whether one, two, or three methyl groups are added to the lysine side chain (*Bhat et al., 2021*). Lysine methylation is catalyzed by a class of enzymes named protein lysine methyltransferases (KMTs) and removed by protein lysine demethylases (*Bhat et al., 2021*). Methylation at H3K36 is an evolutionarily conserved histone modification (*Li et al., 2019*). In humans, mutations in the enzymes that determine H3K36 methylation dynamics are linked to a variety of developmental disorders and cancer (*Li et al., 2019*; *Husmann and Gozani, 2019*; *Bennett et al., 2017*). The state and extent of methylation at H3K36 are synthesized by distinct KMTs, with H3K36me2 generated by four related enzymes (NSD1, NSD2, NSD3, and ASH1L), whereas SETD2 is the only human enzyme in somatic cells that has been reproducibly shown to generate H3K36me3 (*Li et al., 2019*; *Husmann and Gozani, 2019*; *Edmunds et al., 2008*). SETD2 and its cognate mark H3K36me3 generally occupy transcriptionally active regions. Functionally, SETD2 influences core molecular processes such as DNA methylation, RNA processing, DNA repair, and genomic integrity (*Baubec et al., 2015*; *Jha et al., 2014*; *de Almeida et al., 2011*; *Duns et al., 2010*; *Zhu et al., 2014*; *Wen et al., 2014*). Notably, while H3K36me2-generating enzymes like NSD2 and NSD3 promote oncogenesis when mutated or overexpressed (e.g., *Kuo et al., 2011*; *Hudlebusch et al., 2011*; *Jaffe et al., 2013*; *Aytes et al., 2018*; *Yuan et al., 2021*; *Sengupta et al., 2021*), SETD2 is a potent tumor suppressor frequently mutated in clear cell renal cell carcinoma (ccRCC) (*Duns et al., 2010*; *Dalgliesh et al., 2010*) and several other cancers. For example, the SETD2 gene (along with other important tumor suppressors) is located within chromosome 3p, a genomic region that is reported to show loss of heterozygosity in ~90–95% of ccRCC tumors (*Walton et al., 2023*). In ccRCC pathogenesis, ~10–20% of tumors acquire SETD2 mutation on the second intact allele (or more rarely acquire mutations on both alleles), resulting in biallelic loss of SETD2 and H3K36me3 depletion (*Duns et al., 2010*; *Dalgliesh et al., 2010*; *Walton et al., 2023*). Deletions and/or loss-of-function mutations in SETD2 are also detected recurrently in different types of leukemia and solid tumors, including gastroesophageal cancers and lung adenocarcinoma (LUAD), though at lower frequency than seen in ccRCC (*Zhu et al., 2014*; *Foggetti et al., 2021*; *Kadara et al., 2017*; *Walter et al., 2017*; *Mar et al., 2017*).

SETD2 is a large protein with multiple functional regions and domains beyond its ability to catalyze H3K36me3. Thus, the precise role of H3K36me3 in SETD2-associated functions remains unclear. One established function of SETD2-catalyzed H3K36me3 is to allosterically inhibit the ability of the PRC2 complex to synthesize the repressive H3K27me3 chromatin modification (*Cookis et al., 2025*; *Yuan et al., 2011*; *Schmitges et al., 2011*; *Jani et al., 2019*; *Finogenova et al., 2020*; *Chen et al., 2022*). Further, H3K36me3, via its recognition by the PWWP domain of DNMT3b, promotes targeted DNA methylation (*Baubec et al., 2015*). However, beyond H3K27me3 and DNA methylation, potential crosstalk between other epigenetic modifications and SETD2-mediated catalysis of H3K36me3 in the regulation of SETD2 function is relatively unexplored. Here, we test the in vivo role of SETD2 in suppressing KRAS[G12C]-driven tumorigenesis in LUAD, as G12C is the most common KRAS mutation in this tumor type, and identify potential regulatory roles for histone acetylation, particularly H3K27 acetylation, in promoting in vitro methylation of nucleosomes by SETD2.

## Results

### SETD2 deletion accelerates KRAS[G12C]-driven lung cancer pathogenesis in vivo

Several elegant previous in vivo studies utilizing either CRISPR-based or gene knockout approaches demonstrated that loss of SETD2 promotes LUAD tumorigenesis in the canonical KRAS[G12D]-driven LUAD mouse tumor model (*Walter et al., 2017*; *Johnson et al., 2001*; *Jackson et al., 2001*; *Rogers et al., 2018*; *Xie et al., 2023*). As G12C is the most common KRAS oncogenic variant in LUAD (*Campbell et al., 2016*; *Wiesweg et al., 2019*), we used a recently developed KRAS[G12C]-driven LUAD mouse model (named K[c]P) (*Francis et al., 2024*) to test whether the additional loss of SETD2 (named K[c]P;Setd2) impacted cancer pathogenesis in vivo like it does in a G12D oncogenic mutant background (*Figure 1A–C*). In this model, expression of the Kras[G12C] mutant allele and homozygous deletions of Trp53 and Setd2 are induced by intratracheal lavage of adenovirus expressing Cre recombinase

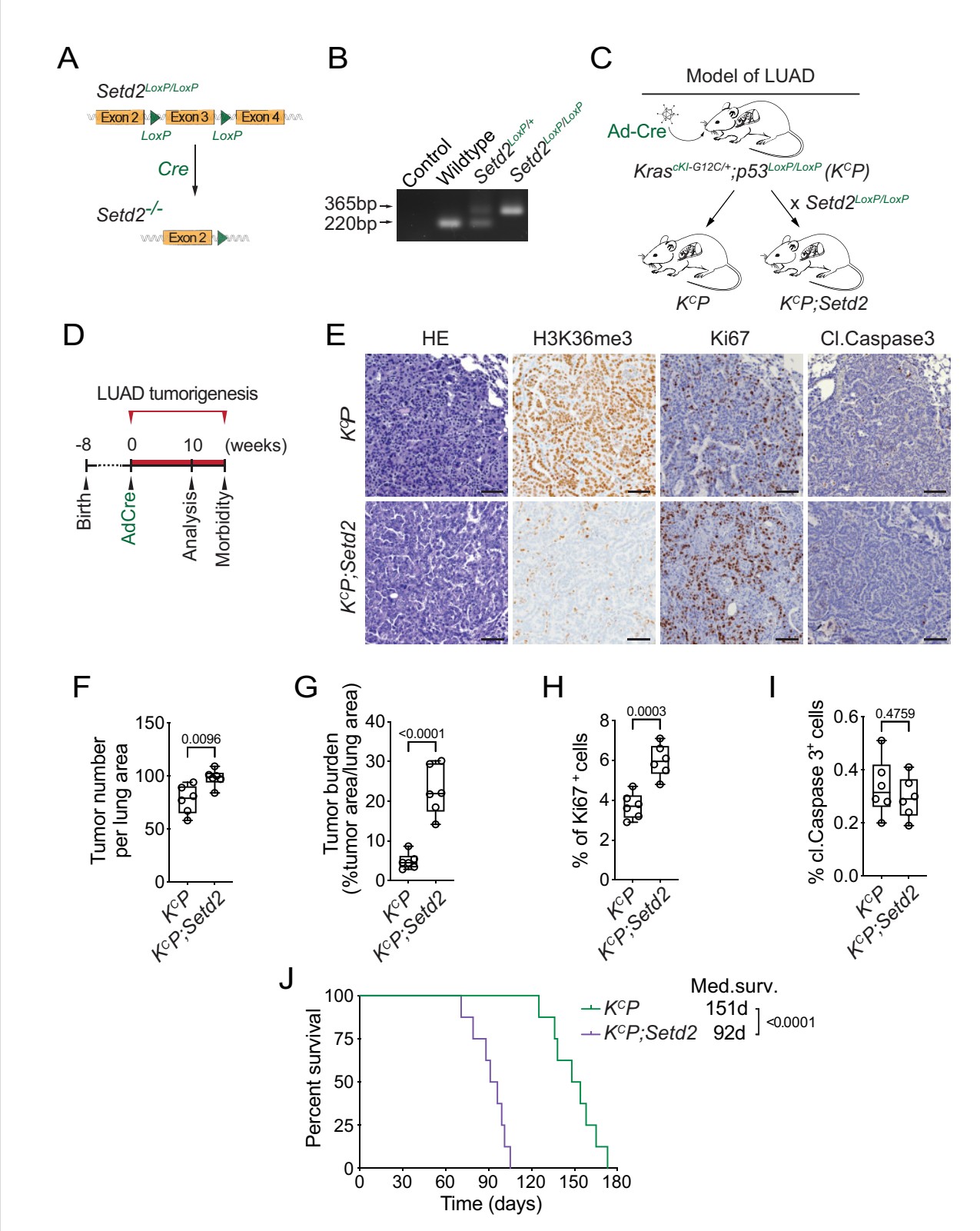

**Figure 1.** SETD2 ablation promotes KRAS[G12C]-driven lung tumorigenesis in vivo. (**A**) Schematic of the Setd2[LoxP/LoxP] conditional allele. In the presence of Cre recombinase, exon 3 is deleted to disrupt Setd2 expression. (**B**) Confirmation of Setd2[LoxP/LoxP] conditional allele by PCR on DNA isolated from mouse tail biopsies from indicated mouse genotypes, expected product sizes are marked. (**C**) Schematic of generation of lung adenocarcinoma (LUAD) model driven by Cre-recombinase inducible conditional oncogenic Kras[G12C] mutation and deletion of p53 (K[c]P) and Setd2 (K[c]P;Setd2). (**D**) Experimental design

*Figure 1 continued on next page*

*Figure 1 continued*

to assess effects of SETD2 ablation on LUAD pathogenesis in K$^C$P model. (**E**) Representative HE and IHC staining with indicated antibodies of lung tumors from K$^C$P and K$^C$P;Setd2 mutant mice at 10 weeks after Ad-Cre induction (n=6/group). H3K36me3 serves as a proxy of SETD2 ablation in tumor cells in K$^C$P;Setd2 mutant mice. p-Values determined by two-tailed unpaired *t*-test; boxes: 25th to 75th percentile, whiskers: min. to max., center line: median; scale bars: 100 µm. (**F–I**) Quantification of tumor number, tumor burden, proliferation (Ki67+) and cell death (cleaved Caspase3+) in K$^C$P and K$^C$P;Setd2 samples as in (**E**). (**J**) Kaplan–Meier survival curves of K$^C$P control (n=8, median survival 151 days) and K$^C$P;Setd2 mutant mice (n=8, median survival 92 days) mutant mice. p-Values determined by the log-rank test.

(Ad-Cre) (*Francis et al., 2024*). Following viral infection, K$^C$P mutant mice develop widespread LUAD with 100% penetrance (see schematic, *Figure 1D*; *Francis et al., 2024*). Consistent with the principal catalytic activity of SETD2 being generation of H3K36me3, Setd2 deletion resulted in loss of H3K36me3 immunohistochemistry (IHC) signal comparing LUAD tissue sections from K$^C$P and K$^C$P;Setd2 mice (*Figure 1E*). Moreover, Setd2 depletion accelerated tumorigenesis as measured by a modest increase in tumor nodule numbers and a fourfold increase in tumor burden (*Figure 1F and G*). Loss of Setd2 also caused an increase in cellular proliferation in tumors, but did not significantly impact apoptosis levels (*Figure 1E, H, I*). Finally, Setd2 depletion resulted in a 40% decline in animal overall median survival time (*Figure 1J*). Thus, as previously observed with oncogenic KRAS$^{G12D}$-based models (*Walter et al., 2017*; *Johnson et al., 2001*; *Jackson et al., 2001*; *Rogers et al., 2018*; *Xie et al., 2023*), Setd2 loss accelerates mutant KRAS$^{G12C}$-driven malignancy in vivo.

## SETD2 in vitro methylation activity on H3K36 methylated nucleosomes

In humans, SETD2 is the only enzyme in somatic cells that generates the canonical epigenetic modification H3K36me3 (*Edmunds et al., 2008*). In contrast, four KMTs (NSD1, NSD2, NSD3, and ASH1L) generate H3K36me2 (*Husmann and Gozani, 2019*). Notably, while SETD2 is tumor suppressive (e.g., see *Figure 1*), the H3K36me2 KMTs are generally oncogenic (*Husmann and Gozani, 2019*). While the specific role of SETD2-catalyzed H3K36me3 in cancer pathogenesis remains unclear, it is intriguing that the di-methyl and tri-methyl states at H3K36 may have profoundly different impacts on tumorigenesis. In this regard, it has been postulated that SETD2 requires pre-existing H3K36me2 to generate the tri-methyl state at K36 (*Li et al., 2019*; *Husmann and Gozani, 2019*). However, previous work demonstrated that the isolated catalytic SET domain of SETD2 methylates unmodified recombinant nucleosomes (rNucs) in vitro with the same efficiency as methyl-lysine analog (MLA)-based H3K$_C$36me2 rNucs (*Li et al., 2009*). While MLA chemistry is generally able to faithfully model native methyl-lysine function (e.g., recognition by reader domains), there are examples where the sulfur moiety compromises functionality (*Seeliger et al., 2012*). Thus, we tested the impact of native methylation installed at H3K36 on the enzymatic activity of the catalytic region of SETD2 that encompasses the SET domain, as well as the pre- and post-SET regions (hereafter named SETD2$_{SET}$; *Figure 2A*).

In vitro methylation assays were performed using recombinant SETD2$_{SET}$, radiolabeled S-adenosyl-methionine (SAM) as the methyl donor, and rNucs that were either unmethylated or harboring me1, me2, or me3 at K36 as substrate. The SETD2$_{SET}$ domain methylated all the rNuc substrates besides the tri-methylated sample (*Figure 2B*). Specifically, the highest signal was observed using unmodified rNucs, with the intensity of the signal decreasing sequentially on H3K36me1 and H3K36me2 rNuc substrates, likely reflecting the reduced capacity of these H3K36 methylated histones to be further methylated (*Figure 2B*). SETD2$_{SET}$ showed no detectable methylation activity on H3K36me3 rNucs, consistent with K36 being 100% saturated with methylation and thus there being no more sites available to be methylated and the specificity of SETD2 for K36 versus other lysine residues on histones (*Figure 2B*). Similar results were observed using an independent quantitative method that measures production of S-adenosyl-homocysteine (SAH)—a key by-product of the methylation reaction (*Figure 2C*). Like with SETD2$_{SET}$, the NSD2$_{SET}$ domain showed the strongest methylation activity on unmodified rNucs compared to H3K36me1 rNucs, and NSD2$_{SET}$ has no activity on H3K36me2/3 rNucs, consistent with NSD2 being a selective H3K36me2-KMT (*Figure 2D*).

To gain insight into the efficiency of conversion between methyl states at K36 for SETD2, we identified methyl-state specific H3K36 antibodies for the three methyl states (*Figure 2E*). We next tested how lysine methyl-state transition on the various H3K36 methylated nucleosome substrates impacts SETD2$_{SET}$ activity. To this end, we performed methylation assays with SETD2$_{SET}$ using non-radiolabeled SAM and detected modification of rNucs using the H3K36me state-specific antibodies (*Figure 2E*).

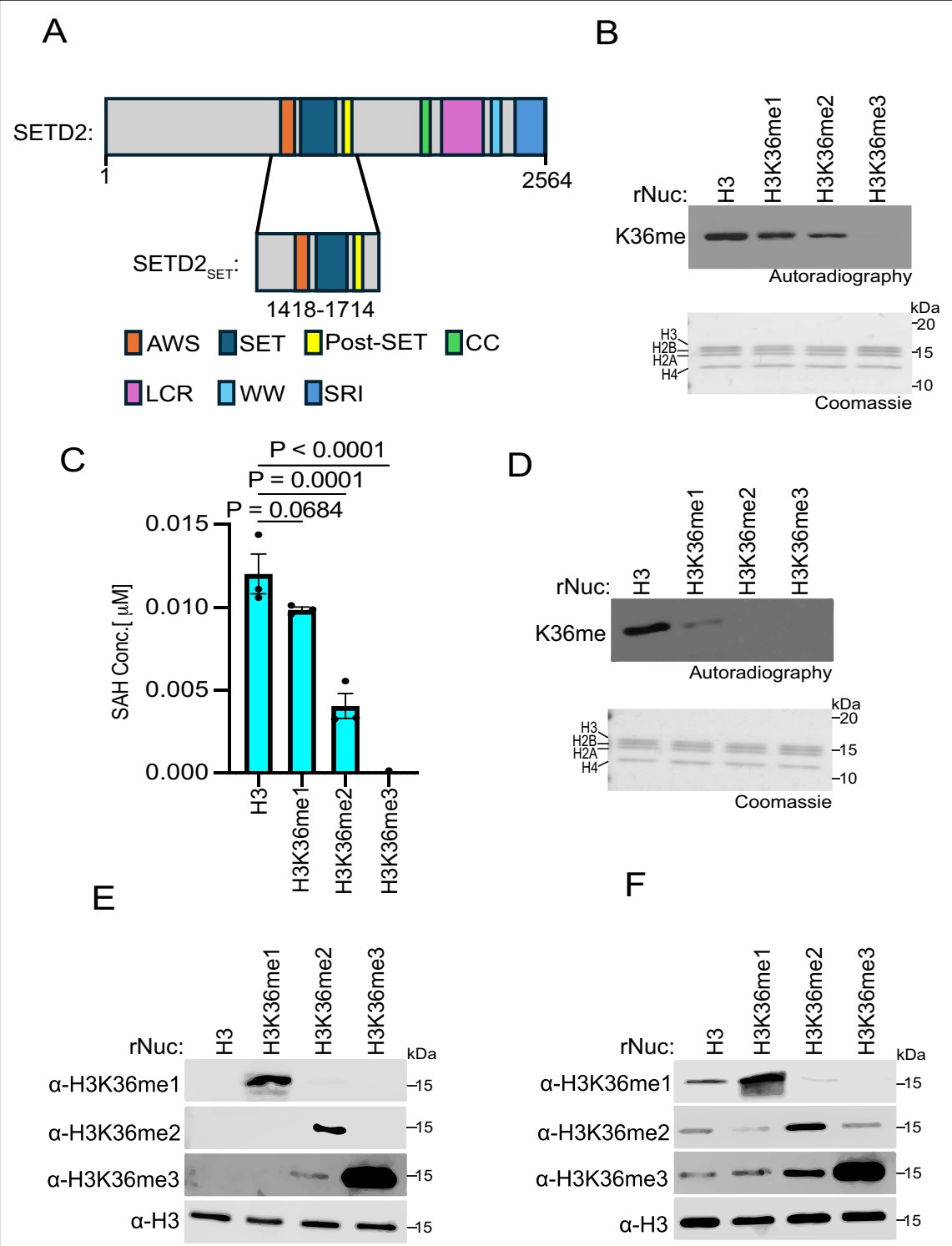

**Figure 2.** SETD2 methylates unmodified, H3K36me1-, and H3K36me2-modified nucleosomes, but not those bearing H3K36me3. (**A**) Schematic of SETD2 domain structure, with the catalytic region used in the biochemical assays (SETD2$_{SET}$) indicated. (**B**) SETD2$_{SET}$ in vitro methylation reactions with radiolabeled $^3$H-SAM on unmodified (H3), H3K36me1, H3K36me2, or H3K36me3 recombinant nucleosomes (rNuc) substrates as indicated. K36me: methylated H3K36. Top, autoradiography; bottom, Coomassie blue staining. (**C**) Methylation (MTase-Glo) assays (see 'Materials and methods') with enzyme and substrates as in (**B**). SAH concentration serves as a measurement of methylation. Activity is normalized to control conditions. Data are means ± SEM from three independent replicates. p-Values determined by one-way ANOVA. (**D**) Methylation reactions as in (**B**) using NSD2$_{SET}$ as the enzyme and the indicated rNucs as substrates. Top, autoradiography; bottom, Coomassie blue staining. (**E**) Western blot analysis with the indicated

*Figure 2 continued on next page*

*Figure 2 continued*

antibodies on the indicated rNucs as in (**B**). (**F**) SETD2$_{SET}$ methylation assays as in (**B**) using non-radiolabeled SAM and methylation detected by Western analyses using the antibodies characterized in (**E**). H3 is shown as a loading control.

The online version of this article includes the following source data for figure 2:

**Source data 1.** Source Data for *Figure 2*.

**Source data 2.** Source Data for *Figure 2* with labels.

Under our reaction conditions (see 'Materials and methods'), SETD2$_{SET}$ generated all three methyl states (me1, me2, and me3) when using unmodified rNucs as substrate (*Figure 2F*). On H3K36me1-rNucs, SETD2$_{SET}$ generated both higher states of methylation at K36 (H3K36me2 and H3K36me3), whereas H3K36me2-rNucs were naturally only converted to the trimethyl state (*Figure 2F*). The conversion to the trimethyl state at K36 was most efficient on H3K36me2-rNucs, which likely reflects the reaction having poor in vitro processivity (*Figure 2F*). These results indicate that the ability of SETD2$_{SET}$ to generate H3K36me3 in vitro does not require pre-existing methylation, consistent with previous studies (*Li et al., 2009*). The data further suggests that, at least in vitro, SETD2$_{SET}$ is agnostic about the state of methylation at H3K36 on nucleosome substrates.

## Specific histone acetylation events enhance SETD2 methylation activity

Histone acetylation promotes transcription and other DNA-templated processes through several mechanisms, such as the recognition of acetyl-lysine by reader proteins and the neutralization of DNA-histone interactions (*Nitsch et al., 2021*; *Ghoneim et al., 2021*; *Jain et al., 2023*). The combined consequences of such activities decondense chromatin to increase accessibility to the underlying DNA. In addition, neutralizing the positive charge on the lysine side chain facilitates accessibility of KMTs to the unstructured histone tails by disrupting tail-DNA interactions, as was previously shown for the MLL1 KMT complex (*Ghoneim et al., 2021*; *Jain et al., 2023*; *Marunde et al., 2024*; *Fox et al., 2024*). Given the relatively restrictive location of K36 of H3 near the globular domain of the nucleosome, we speculated that histone acetylation might render the residue more accessible to methylation by SETD2. To test this idea, methylation assays with SETD2$_{SET}$ were performed using substrates that were either unmodified rNucs or rNucs with tetra-acetylated lysine residues on the N-terminus tails of H2A, H3, or H4 (see schematic, *Figure 3A*). SETD2$_{SET}$ methylation activity as determined by incorporation of radiolabeled SAM or SAH production was enhanced on tetra-acetylated rNucs compared to control rNucs irrespective of the acetylated histone tail (*Figure 3B and C*). A similar trend was observed in methylation assays using the NSD2$_{SET}$ domain, though H3 tetra-acetylation seemed to have the biggest impact compared to acetylation of H2A and H4 (*Figure 3D*). In contrast, methylation assays using hDOT1L, the KMT responsible for H3K79 methylation—a PTM situated in the nucleosome globular domain—were not enhanced by acetylation of the various histone tails (*Figure 3E*). Collectively, these data suggest that a high degree of acetylation may increase the accessibility of H3K36 to the enzymes that methylate this residue.

We next explored whether acetylation of individual residues on the H3 tail could facilitate SETD2 activity, and if so, whether such an effect depends on the specific acetylation site. To this end, methylation assays with SETD2$_{SET}$ were performed using substrates that were either unmodified rNucs or rNucs mono-acetylated at K4, K9, K14, K18, K23, or K27 on the N-terminal tail of H3 (see schematic, *Figure 4A*). Methylation activity, determined by incorporation of radiolabeled SAM (*Figure 4B*) or SAH production (*Figure 4C*), indicated that three different single acetylation sites (K14, K23, and K27) enhanced SETD2$_{SET}$ methylation activity on nucleosomes, with the impact likely H3K27ac>H3K14ac>H3K23ac (*Figure 4B and C*). While understanding the molecular basis for the acetylation-mediated enhancement in SETD2 activity will likely require structural investigation, we reasoned that given the proximity and relationship between H3K27 and H3K36 (*Cookis et al., 2025*; *Li et al., 2021*), acetylation at K27 might impact the interaction between SETD2$_{SET}$ and nucleosomes. Consistent with this, SETD2$_{SET}$ was subtly more efficiently pulled down when using H3K27ac rNucs as bait compared to unmodified rNucs and the ability of SETD2$_{SET}$ to form a complex with recombinant mono-nucleosomes was subtly enhanced in the presence of K27 acetylation (*Figure 4D and E*). These data suggest K27ac may directly or indirectly minorly influence the interaction between SETD2$_{SET}$ and its substrate site in the context of nucleosomes (see 'Discussion').

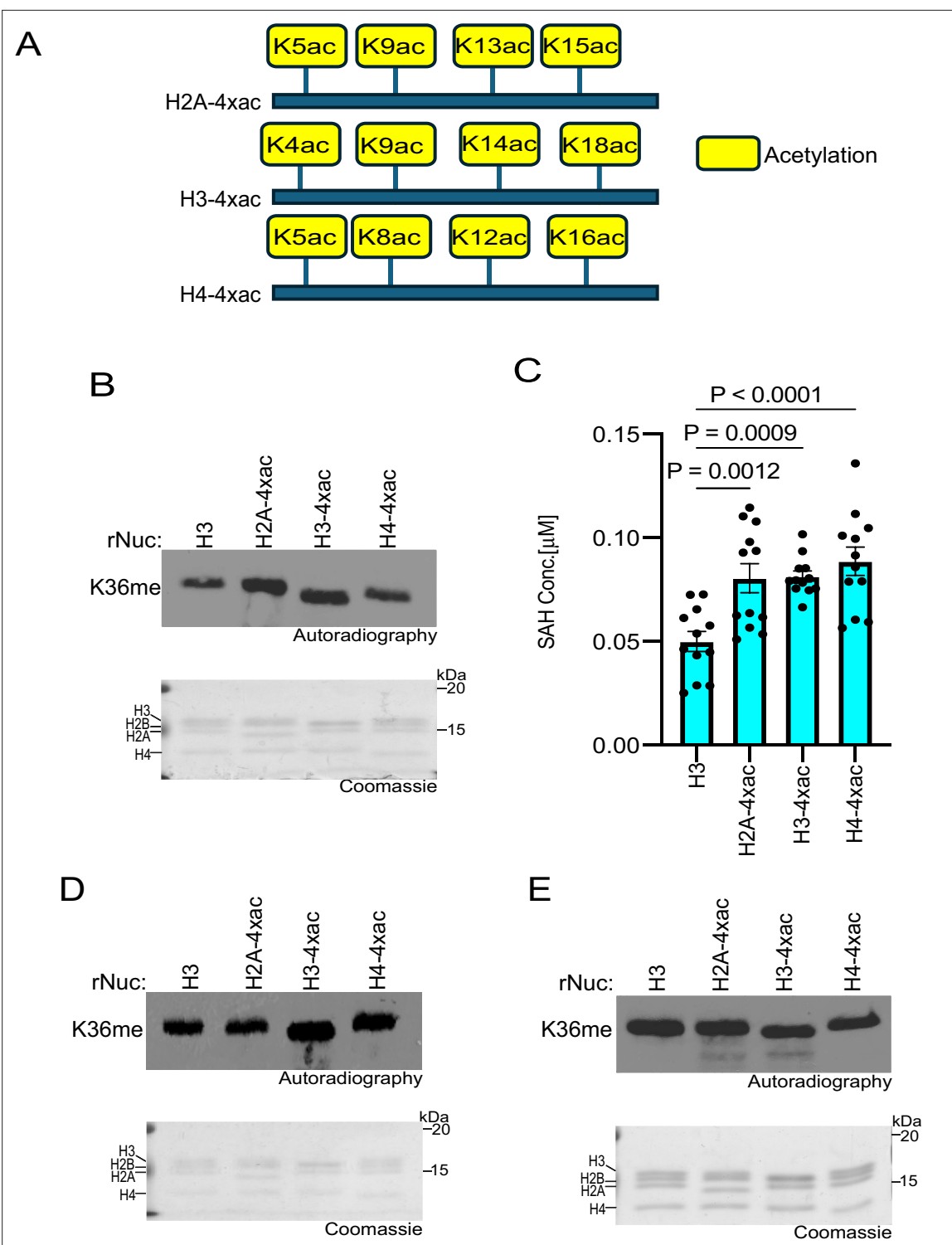

**Figure 3.** Histone poly-acetylation enhances SETD2$_{SET}$ activity. (**A**) Schematic showing the tetra-acetylated rNuc tested in the study. (**B**) In vitro methylation reactions with SETD2$_{SET}$ as in *Figure 2B* using the indicated tetra-acetylated rNucs. Top, autoradiography; bottom, Coomassie blue staining. (**C**) MTase-Glo assays as in *Figure 2C* using the indicated tetra-acetylated rNucs. Data are means ± SEM from 12 independent replicates. p-Values determined by one-way ANOVA. (**D**) Methylation reactions as in (**B**) using NSD2$_{SET}$ as enzyme. Top, autoradiography; bottom, Coomassie blue staining. (**E**) Methylation reactions as in (**B**) using DOT1L as enzyme. Top, autoradiography; bottom, Coomassie blue staining.

The online version of this article includes the following source data for figure 3:

**Source data 1.** Source Data for *Figure 3*.

**Source data 2.** Source Data for *Figure 3* with labels.

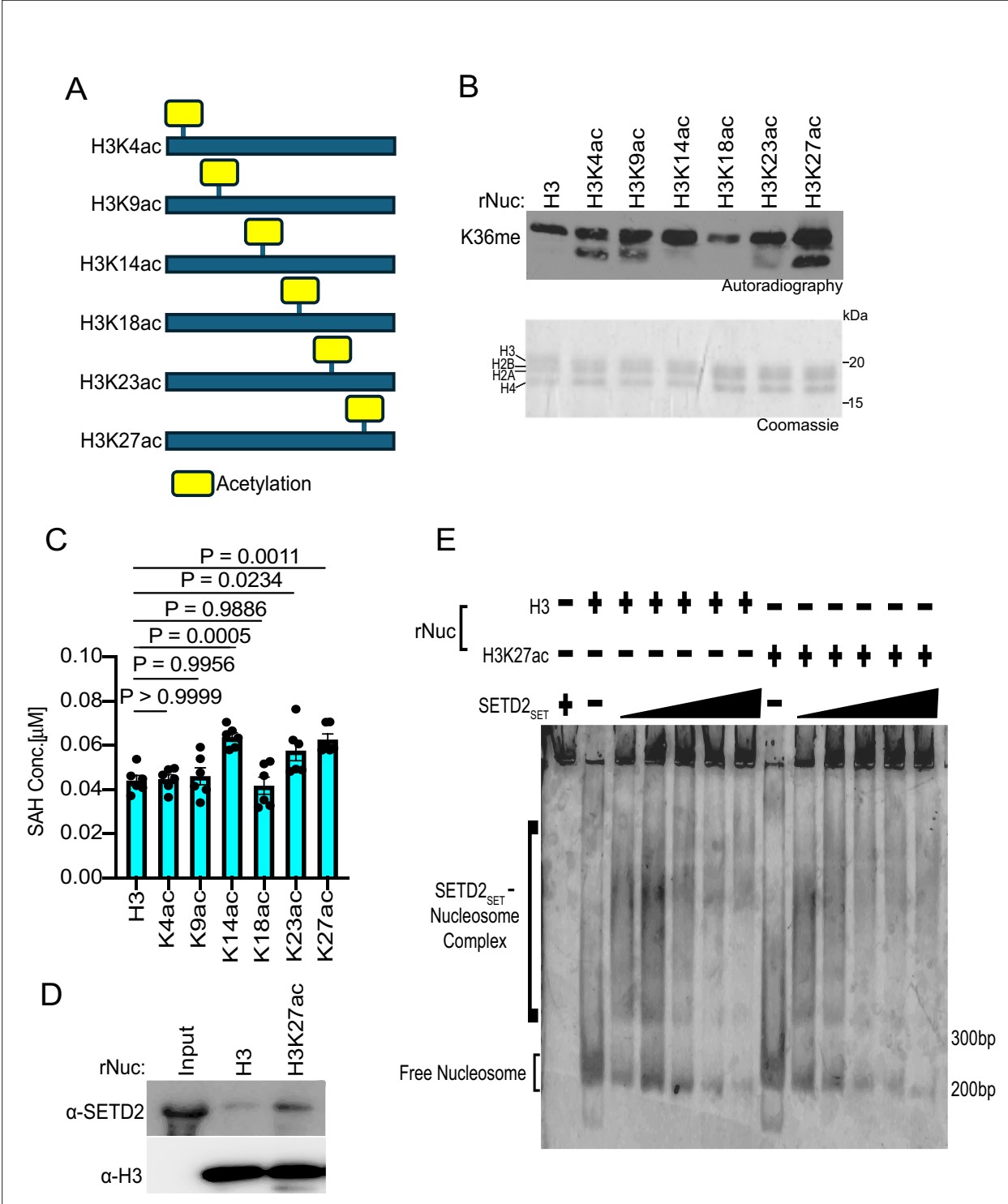

**Figure 4.** H3K27 acetylation enhances SETD2$_{SET}$ binding and activity. (**A**) Schematic showing the different histone H3 acetylated rNuc tested in the study. (**B**) In vitro methylation reactions with SETD2$_{SET}$ as in *Figure 2B* using the indicated acetylated rNucs. Top, autoradiography; bottom, Coomassie blue staining. (**C**) MTase-Glo assays as in *Figure 2C* using the indicated acetylated rNucs. Data are means ± SEM from six independent replicates. p-Values determined by one-way ANOVA. (**D**) Nucleosome pulldown assay using biotinylated unmodified or H3K27ac nucleosomes as bait to assess binding to GST-SETD2$_{SET}$. Bound protein was detected by Western blotting with indicated antibodies. (**E**) Electrophoretic mobility shift assay (EMSA) with increasing concentrations of SETD2$_{SET}$ (0–5 μM) incubated with the indicated rNuc (250 nM) and SETD2$_{SET}$ binding to rNuc detected by staining DNA with Sybr Gold.

The online version of this article includes the following source data for figure 4:

*Figure 4 continued on next page*

*Figure 4 continued*

**Source data 1.** Source Data for *Figure 4*.

**Source data 2.** Source Data for *Figure 4* with labels.

## Histone ubiquitination does not impact SETD2 methylation activity in vitro

We next investigated how ubiquitination at various histone residues influences SETD2 activity. While methylation at H3K36 directly blocks a key allosteric interaction between the PRC2 complex and unmodified H3K36, H3K27me3 indirectly inhibits H3K36 KMTs. Specifically, H3K27me3 recruits the PRC1 complex, which catalyzes the ubiquitination of H2AK119 (H2AK119ub) (*de Napoles et al., 2004*). Nucleosomes harboring H2AK119ub antagonize H3K36 enzymes like NSD2 by structurally preventing their proper association with the nucleosome (*Li et al., 2021*). However, whether H2AK119ub and/or other histone ubiquitination sites impact SETD2 activity is unclear, although studies in yeast have shown the SETD2 homolog, Set2, is influenced by H2BK120ub (*Bilokapic and Halic, 2019*). Thus, we tested SETD2$_{SET}$ activity against a library of ubiquitinated nucleosomes as substrates (see schematic, *Figure 5A*). Surprisingly, assaying methylation by three different methods (utilizing radiolabeled SAM, detecting the different methylation states by Western, and measuring SAH generation), histone ubiquitination did not impact SETD2 activity (*Figure 5B–D*; note that the H3K36me2 antibody is blocked by H3K14 and H3K18 ubiquitination). In contrast, as expected, H2AK119ub specifically reduced NSD2 activity (*Figure 5E*). Collectively, these findings indicate that the isolated catalytic domain of SETD2, unlike for NSD2, is not influenced by histone ubiquitination, suggesting potential functional differences in how H3K36 enzymes interact with the ubiquitinated nucleosome landscape.

## Discussion

SETD2, via catalytic and non-catalytic mechanisms, regulates fundamental nuclear processes. Mutations in SETD2 cause Sotos-like syndrome, an overgrowth disorder, and loss of SETD2 commonly occurs and is thought to be tumor suppressive in ccRCC and many other malignancies (*Husmann and Gozani, 2019*; *Luscan et al., 2014*). Indeed, SETD2 was identified as one of the most mutated genes in a saturation analysis of 21 cancer types (*Lawrence et al., 2014*). Consistent with these observations and previous lung cancer mouse modeling studies (*Walter et al., 2017*; *Rogers et al., 2018*; *Xie et al., 2023*), we find that Setd2 ablation strongly accelerates LUAD malignant progression and lethality in a KRAS$^{G12C}$-driven mouse tumor model (see *Figure 1*). While these studies demonstrate that CRISPR-mediated endogenous depletion of SETD2 or homozygous genetic deletion of Setd2 promote tumorigenesis, future work will help elucidate the specific contributions of H3K36me3 synthesis and other SETD2 functional domains in tumor suppression. In this context, EZM0414, a clinical-grade SETD2 inhibitor, was being evaluated in a phase I clinical trial as a therapeutic in relapsed or refractory multiple myeloma and diffuse large B cell lymphoma, suggesting that at least in these cancer types, H3K36me3 generation may be oncogenic (NCT05121103; note this trial was terminated). Whether SETD2-mediated catalysis of H3K36me3 is tumor suppressive or oncogenic in a tumor context-dependent manner—and how this activity relates to SETD2's other functions—are potentially important questions for future investigation.

Consistent with previous work, our biochemical analyses using native designer nucleosomes indicate that—at least in isolation—the catalytic domain of SETD2 does not exhibit a preference for substrates bearing mono- or di-methylation at K36 compared to the unmodified state (see *Figure 2*; *Li et al., 2009*). We speculate that physiologically SETD2 uses all three lower states of methylation at H3K36me (me0-me2) to generate endogenous H3K36me3, with the specific precursor state determined by the underlying biology and genomic context. Our analyses also uncovered an unanticipated regulatory role for histone acetylation in activating SETD2 in vitro methylation activity on nucleosomes. Why some acetylation sites increased SETD2 activity whereas others did not is presently unclear. One possibility is that certain acetylation events, for example, H3K27ac, may alter histone-DNA interactions in a manner that renders H3K36 more accessible to the catalytic pocket of SETD2. Additionally, our data suggests that SETD2 may interact more strongly with H3K27ac nucleosomes compared to unmodified nucleosomes, which would promote higher levels of methylation

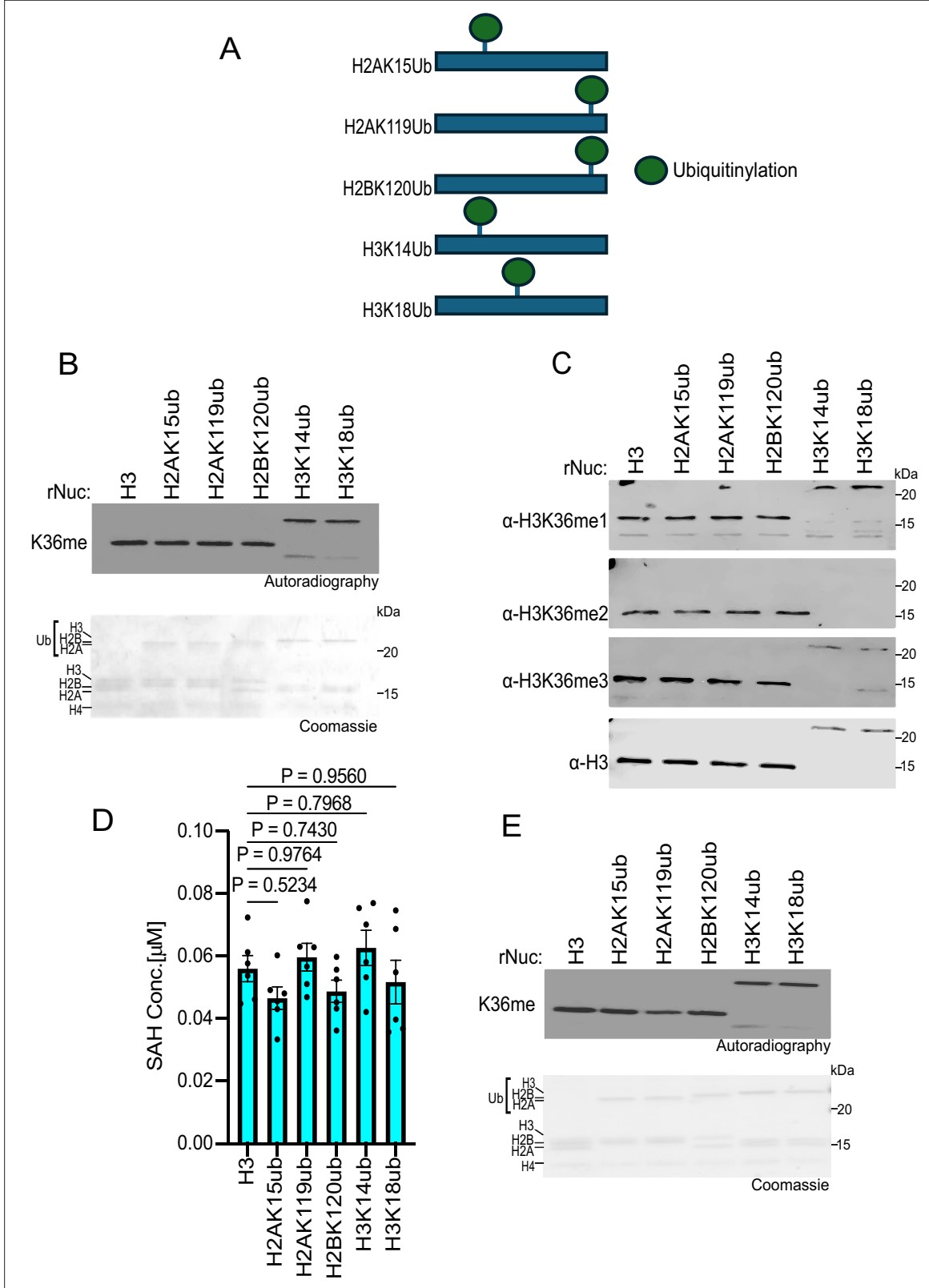

**Figure 5.** Histone ubiquitination does not affect SETD2$_{SET}$ activity. (**A**) Schematic showing the different histone ubiquitinated rNuc tested in the study. (**B**) In vitro methylation reactions with SETD2$_{SET}$ as in *Figure 2B* using the indicated ubiquitinated rNucs. Top, autoradiography; bottom, Coomassie blue staining. (**C**) SETD2$_{SET}$ methylation assays as in *Figure 2F* using non-radiolabeled SAM and methylation detected by Western analysis using the indicated antibodies. H3 is shown as a loading control. Note that H3K14ub and H3K18ub interfere with the αH3K36me2 antibody from recognizing

*Figure 5 continued on next page*

*Figure 5 continued*

H3K36me2. (**D**) MTase-Glo assays as in *Figure 2C* using the indicated rNucs. Data are means ± SEM from six independent replicates. p-Values determined by one-way ANOVA. (**E**) Methylation reactions as in (**B**) using NSD2$_{SET}$ as enzyme. Top, autoradiography; bottom, Coomassie blue staining.

The online version of this article includes the following source data for figure 5:

**Source data 1.** Source Data for *Figure 5*.

**Source data 2.** Source Data for *Figure 5* with labels.

---

(*Figure 4*). Finally, we observed a divergence between SETD2 and H3K36me2-KMTs such as NSD2, in that SETD2 is neither inhibited by H2AK119ub nor regulated by H2BK120ub (*Figure 5*). These data are consistent with recent cryo-EM-based structural studies, which provide a molecular rationale for why NSD2, but not SETD2, would be impacted by a large modification within the C-terminal region of H2A (*Li et al., 2021*; *Liu et al., 2021*; *Markert et al., 2025*). Future functional and epigenomic studies will be crucial for understanding how epigenetic modifications such as acetylation and ubiquitination regulate SETD2 activity in physiological and pathological settings.

## Materials and methods

### Protein expression and purification

The SETD2$_{SET}$ (aa1418-1714, NCBI sequence: NC_000003.12), NSD2$_{SET}$ (aa959-1365, NCBI sequence: NC_000004.12), and DOT1L (aa1-416, NCBI sequence: NC_000019.10) were cloned into pGEX-6P-1 separately. *Escherichia coli* Rossetta cells were transformed with the respective expression vectors and cultured in LB medium (10 g/L tryptone, 5 g/L yeast extract, and 10 g/L NaCl) supplemented with 0.1 mM isopropyl 1-thio-β-D-galactopyranoside (IPTG, Sigma) at 18°C for 16–20 h. Cells were lysed using a sonicator, lysates were cleared by centrifugation at 12,000 rpm for 20 min and the supernatants were incubated with Glutathione Sepharose (GE Healthcare #17-0756-01); bound proteins were washed and eluted in 10 mM reduced glutathione (Sigma #G4251-25G). Protein concentrations were measured using Pierce Coomassie Plus Assay (Thermo #23236).

For expression and purification of GST-SETD2 (1345–1711) used in *Figure 4D*, BL21.DE3(pLysS) *E. coli* transformed with a recombinant expression plasmid encoding GST-tagged human SETD2 catalytic domain (residues 1345–1711) was grown in LB supplemented with 50 µg/mL carbenicillin at 37°C until OD600 reached ~0.6. Cultures were then transferred to 16°C and induced with 1 mM IPTG (Sigma) overnight and harvested by centrifugation, flash frozen in liquid nitrogen, and stored at –20°C until use. Thawed cell pellets were resuspended in lysis buffer (50 mM HEPES pH 7.5, 150 mM NaCl, 1 mM dithiothreitol (DTT), 1 mM PMSF) supplemented with 250 U of Pierce Universal Nuclease (Thermo Fisher) and 1 mg/mL lysozyme (Sigma) and incubated at 37°C for 10 min. Cells were then lysed by sonication (5 × 30 s, 40% cycle, 40% power) and lysates clarified by centrifugation before application to glutathione agarose beads (Pierce) pre-equilibrated in wash buffer (50 mM HEPES pH 7.5, 150 mM NaCl, 1 mM DTT, 1 mM PMSF). Following three sequential washes with wash buffer, proteins were eluted in ~1 mL fractions with wash buffer supplemented with 10 mM glutathione. Fractions containing purified GST-SETD2 were pooled, concentrated by centrifugation filtration (EMD Millipore, MWCO 30 kDa). Glycerol was added to 20% final concentration, aliquoted, and stored at –80°C until use (*Hacker et al., 2016*).

### In vitro methylation reactions

The methylation reactions on nucleosomes were performed similarly to those previously described (*Wang et al., 2020*). Briefly, 350 nM recombinant enzymes were mixed with 500 nM mononucleosome (EpiCypher, H3-unmodified #16-0006, H3K36me1 #16-0322, H3K36me2 #16-0319, H3K36me3#16-0390, H2A-4xac#16-0376, H3-4xac#16-0336, H4-4xac#16-0313, H3K4ac#16-0342, H3K9ac#16-0314, H3K14ac#16-0343, H3K18ac#16-0372, H3K23ac#16-0364, H3K27ac#16-0365, H2AK15ub#16-0399, H2BK119ub#16-0395, H2AK120ub#16-0396, H3K14ub#16-0398, H3K18ub#16-0401) in reaction buffer (containing 250 mM Tris pH 8.0, 100 mM KCl, 25 mM MgCl$_2$, and 50% glycerol), after adding 20 µM SAM, the mixture was incubated at 30°C for 3 h.

## MTase-Glo methyltransferase assay

The MTase-Glo methyltransferase assay kit (Promega #V7602) was used to measure enzymatic activity in the presence of different substrates. Specifically, we established an assay in 10 µL reaction mix containing 35 nM KMT enzyme, 4 µM SAM, 50 nM mononucleosomes (EpiCypher), and MTase-Glo reagent (1×) in reaction buffer (containing 250 mM Tris pH 8.0, 100 mM KCl, 25 mM $MgCl_2$, and 50% glycerol) arrayed in a white 384-well microplate (Corning #CLS3574). Each independent biochemical reaction was performed in triplicate and incubated for 3 h at 30°C. Subsequently, 10 µL of MTase-Glo detection solution was added and incubated for 1 h at room temperature. Reactions were detected by luminescence.

## Western blot analyses

For western blot analysis, protein samples were resolved by SDS–PAGE and transferred to a PVDF membrane. The following antibodies were used (at the indicated dilutions): H3K36me1 (Abclonal #A11141, 1:1000), H3K36me2 (Thermo Fisher #701767, 1:1000), H3K36me3 (EpiCypher #13-0058, 1:1000), H3 (EpiCypher #13-0001, 1:10,000).

## Animal models

Kras$^{cKI-G12C}$ and Trp53$^{loxP/loxP}$ mutant mice have been described before (*Francis et al., 2024*; *Jonkers et al., 2001*). Conditional Setd2$^{loxP/loxP}$ mouse strain was obtained from Shanghai Model Organisms Center, Inc (Cat# NM-CKO-190069). Briefly, the Setd2$^{loxP/loxP}$ targeted knockin sequence includes the Neo-LacZ cassette flanked by Frt sites and exon 3 sequence flanked by LoxP sites. Founder mice were crossed with Rosa26-FlpO deleter strain (*Raymond and Soriano, 2007*) to generate Setd2$^{loxP/loxP}$ conditional allele. Confirmation of Setd2$^{loxP/loxP}$ conditional allele targeting was performed by PCR on DNA isolated from mouse tail biopsies and the following primers: forward: AGCTGACCTGAT TTCTCCTTTAG; reverse: AACAGCTGAGAGTGACCATGAG. Mice were maintained on a mixed C57BL/6;FVB strain background, and we systematically used littermates as controls in all the experiments. Both male and female animals were used in the experiments, and no sex differences were noted. In all experiments, animals were numbered, and experiments were conducted in a blinded fashion. After data collection, genotypes were revealed, and animals were assigned to groups for analysis. None of the mice with the appropriate genotype were excluded from this study or used in any other experiments. All mice were co-housed with littermates (2–5 per cage) in a pathogen-free facility with standard controlled temperature of 72°F, with a humidity of 30–70%, and a light cycle of 12 h on/12 h off set from 7 am to 7 pm and with unrestricted access to standard food and water under the supervision of veterinarians, in an AALAC-accredited animal facility at the University of Texas M.D. Anderson Cancer Center (MDACC). Mouse handling and care followed the NIH Guide for Care and Use of Laboratory Animals. All animal procedures followed the guidelines of and were approved by the MDACC Institutional Animal Care and Use Committee (IACUC protocol 00001636, PI: Mazur).

To evaluate the effects of SETD2 ablation on the development and progression of LUAD, we utilized Kras$^{cKI-G12C/+}$, Trp53$^{loxP/loxP}$ (K$^C$P), and Kras$^{cKI-G12C/+}$, Trp53$^{loxP/loxP}$, Setd2$^{loxP/loxP}$ (K$^C$P;Setd2). To generate tumors in the lungs of K$^C$P;Setd2 and control K$^C$P mutant mice, we used replication-deficient adenoviruses expressing Cre-recombinase (Ad-Cre) as previously described (*Liu et al., 2019*). Briefly, 8-week-old mice were anesthetized by continuous gaseous infusion of 2% isoflurane for at least 10 min using a veterinary anesthesia system. Ad-Cre was delivered to the lungs by intratracheal lavage. Prior to administration, the virus was precipitated with calcium phosphate to improve the delivery of Cre by increasing the efficiency of viral infection of the lung epithelium. Mice were treated with one dose of 5 × 10$^6$ PFU of Ad-Cre. Mice were analyzed for tumor formation and progression at indicated timepoints after viral infection. The survival endpoint was determined by overall health criteria scoring. Mouse health status and weight were checked daily.

## Histology and immunohistochemistry

Tissue specimens were fixed in 4% buffered formalin for 24 h and stored in 70% ethanol until paraffin embedding. 3 µm sections were stained with hematoxylin and eosin (HE) or used for immunostaining studies. The following antibodies were used (at the indicated dilutions): cleaved Caspase 3 (CST #9664, 1:200), Ki67 (BD Bioscience #550609, 1:1000), and H3K36me3 (CST #4909, 1:1000). Immunohistochemistry (IHC) was performed on formalin-fixed, paraffin-embedded tissue (FFPE) sections

using a biotin-avidin HRP conjugate method (Vectastain ABC-HRP kit, #PK4000) as described before (*Park et al., 2024*). Sections were developed with DAB and counterstained with hematoxylin. Pictures were taken using a PreciPoint M8 microscope equipped with the PointView software and quantified using ImageJ software (v1.53k, RRID:SCR_003070) and QuPath (v0.5.1, RRID:SCR_018257).

## Nucleosome pulldown assays

2.5 pmol of GST-SETD2 was added to nucleosome binding buffer (50 mM Tris-Cl pH 7.6, 300 mM NaCl, 0.1% NP-40, 0.5% bovine serum albumin, 10% glycerol) to a final volume of 25 µL. 12.5 pmol of biotinylated nucleosome was added and rotated overnight at 4°C. 1 µL of streptavidin Dynabeads (Fisher) was equilibrated in nucleosome binding buffer and resuspended to a final volume of 7.5 µL. Resuspended beads were then added to the nucleosome-enzyme mixture and rotated at 4°C for 1 h. Beads were pelleted on a magnet, the supernatant (unbound) fraction was collected, and the beads were washed with 200 µL of nucleosome binding buffer three times. Following the final wash, beads were resuspended in 15 µL of a 1× SDS-PAGE loading dye and stored at 4°C until immunoblotting.

## Electromobility shift assay (EMSA)

Nucleosomes were incubated with recombinant SETD2$_{SET}$ in EMSA buffer (30 mM Tris-HCl pH 7.5, 100 mM KCl, 25 mM MgCl$_2$, 6 mM DTT, 0.0075% Tween 20, 12% glycerol) for 15 min at room temperature and analyzed by native 0.2X TBE-PAGE. Each reaction contained 250 nM of nucleosome with increasing concentrations of domain (0, 1, 2, 3, 4, 5 µM). Gels were stained with Sybr Gold (Thermo #S11494).

## Quantification and statistical analysis

Refer to the figure legends or the experimental details for a description of sample size (n) and statistical details. All values for n are for individual mice or individual samples. Sample sizes were chosen based on previous experience with given experiments. Differences were statistically analyzed by unpaired two-tailed *t*-test and one-way ANOVA with Dunnett's test for multiple comparisons as indicated.

## Data availability

The reagents generated in this study will be available from the lead contact upon request with a completed material transfer agreement.

# Acknowledgements

This work was supported in part by grants from the NIH to OG (R35 GM139569), OG and PKM (R01 CA272844, R01 CA278940), and PKM (R01 CA236949, R01 CA266280, R01 CA272843). PKM is also supported by DoD PRCRP Career Development Award (CA181486), CPRIT IIRA (RP220391), and CPRIT Scholar in Cancer Research (RR160078). RM is supported by a DARE fellowship. This work was also supported in part by grants from the NIH to BDS (R35 GM126900) and JMD (R35 GM152103). GCF was supported by a predoctoral training grant from the National Institute for General Medical Sciences (T32 GM135128).

# Additional information

### Competing interests

Brian D Strahl: B.D.S. is a board member, co-scientific founders, and shareholders of EpiCypher, Inc. Pawel K Mazur: P.K.M. is a consultant and stockholder of Ikena Oncology, Inc and Alternative bio, Inc. Or Gozani: O.G. is a co-scientific founder and shareholders of K36 Therapeutics, Inc and Alternative Bio, Inc and a board member, co-scientific founders, and shareholders of EpiCypher, Inc. The other authors declare that no competing interests exist.

## Funding

| Funder | Grant reference number | Author |
| --- | --- | --- |
| National Cancer Institute | RO1CA272844 | Pawel K Mazur<br>Or Gozani |
| National Cancer Institute | RO1CA278940 | Pawel K Mazur<br>Or Gozani |
| National Institute of General Medical Sciences | R35GM139569 | Or Gozani |
| National Cancer Institute | RO1CA236949 | Pawel K Mazur |
| National Cancer Institute | RO1CA266280 | Pawel K Mazur |
| National Cancer Institute | RO1CA272843 | Pawel K Mazur |
| DOD Peer Reviewed Cancer Research Program | CA181486 | Pawel K Mazur |
| National Institute of General Medical Sciences | R35GM126900 | Brian D Strahl |
| National Institute of General Medical Sciences | R35GM152103 | Jill M Dowen |
| National Institute of General Medical Sciences | T32GM135128 | Geoffrey C Fox |
| DOD Peer Reviewed Cancer Research Program | RP220391 | Pawel K Mazur |
| CPRIT Scholar in Cancer Research | RR160078 | Pawel K Mazur |

The funders had no role in study design, data collection and interpretation, or the decision to submit the work for publication.

## Author contributions

Ricardo J Mack, Conceptualization, Investigation, Writing – original draft; Natasha M Flores, Geoffrey C Fox, Hanyang Dong, Metehan Cebeci, Simone Hausmann, Tourkian Chasan, Investigation; Jill M Dowen, Supervision, Writing – review and editing; Brian D Strahl, Pawel K Mazur, Conceptualization, Supervision, Writing – review and editing; Or Gozani, Conceptualization, Supervision, Writing – original draft, Writing – review and editing

## Author ORCIDs

Ricardo J Mack ![ORCID] https://orcid.org/0000-0003-1078-9888
Geoffrey C Fox ![ORCID] https://orcid.org/0000-0001-5898-0847
Tourkian Chasan ![ORCID] https://orcid.org/0000-0001-9108-2205
Brian D Strahl ![ORCID] https://orcid.org/0000-0002-4947-6259
Or Gozani ![ORCID] https://orcid.org/0000-0002-1365-4463

## Ethics

Mouse handling and care followed the NIH Guide for Care and Use of Laboratory Animals. All animal procedures followed the guidelines of and were approved by the MDACC Institutional Animal Care and Use Committee (IACUC protocol 00001636, PI: Mazur).

Reviewer #1 (Public review): https://doi.org/10.7554/eLife.107451.3.sa1
Reviewer #2 (Public review): https://doi.org/10.7554/eLife.107451.3.sa2
Author response https://doi.org/10.7554/eLife.107451.3.sa3

# Additional files

## Supplementary files

MDAR checklist

### Data availability

All data generated or analysed during this study are included in the manuscript and supporting files; source data files have been provided.

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
