## [Editor Report · eLife Assessment]

This is a **fundamental** study providing molecular insight into how cross-talk between histone modifications regulates the histone H3K36 methyltransferase SETD2. The article contains excellent quality data, and the conclusions are **convincing** and justified. This work will be of interest to many biochemists working in the field of chromatin biology and epigenetics.

---

## [Referee Report · Reviewer #1 (Public review)]

Summary:

In this manuscript, Mack and colleagues investigate the role of posttranslational modifications, including lysine acetylation and ubiquitination, in methyltransferase activity of SETD2 and show that this enzyme functions as a tumor suppressor in a KRASG12C-driven lung adenocarcinoma. In contrast to H3K36me2-specific oncogenic methyltransferases, the deletion of SETD2, which is capable of H3K36 trimethylation, increases lethality in a KRASG12C-driven lung adenocarcinoma mouse tumor model. In vitro, the authors demonstrate that polyacetylation of histone H3, particularly of H3K27, H3K14 and H3K23, promotes the catalytic activity of SETD2, whereas ubiquitination of H2A and H2B has no effect.

Strengths:

Overall, this is a well-designed study that addresses an important biological question regarding the functioning of the essential chromatin component. The manuscript contains excellent quality data, and the conclusions are convincing and justified. This work will be of interest to many biochemists working in the field of chromatin biology and epigenetics.

Comments on revisions:

All previous comments are well addressed, and I enthusiastically support publication.

---

## [Referee Report · Reviewer #2 (Public review)]

Summary:

Human histone H3K36 methyltransferase Setd2 has been previously shown to be a tumor suppressor in lung and pancreatic cancer. In this manuscript by Mack et al., the authors first use a mouse KRASG12D-driven lung cancer model to confirm in vivo that Setd2 depletion exacerbates tumorigenesis. They then investigate the enzymatic regulation of the Setd2 SET domain in vitro, demonstrating that H2A, H3, or H4 acetylation stimulates Setd2-SET activity, with specific enhancement by mono-acetylation at H3K14ac or H3K27ac. In contrast, histone ubiquitination has no effect. The authors propose that H3K27ac may regulate Setd2-SET activity by facilitating its binding to nucleosomes. This work provides insight into how cross-talk between histone modifications regulates Setd2 function.

Comments on revisions:

(1) Regarding New Figure 2F lane 1, please reference PMID: 33972509 Fig 4D bottom. Setd2-SET is a well-known robust K36 trimethylase. Why, under the authors' conditions, do WT nucleosomes show a significant amount of K36me1 and K36me2 accumulation, whereas K36me3 is not as pronounced? As a comparison, the authors should also report the evidence for the efficiency of each chemical modification that generates K36 methylation mimic.

(2) The bottom panel of Figure 2B does not match the top one; the number of repeats should be indicated in the figure legends.

(3) In Figure 4E, the differences between Setd2-bound WT and acetylated nucleosomes are minimal, as judged by both the decreasing trend of unbound nucleosomes and the increasing trend of bound fractions. This experiment needs to be quantified based on multiple repeats.

---

## [Author Response]

The following is the authors’ response to the original reviews.

**Reviewer #1 (Public review):**
(1) Labels should be added in the Figures and should be uniform across all Figures (some are distorted).

We thank the Reviewer for pointing out this issue. As requested, labels have been edited to ensure they are legible and are consistent in font, size, and style.

**Reviewer #2 (Public review):**
(1) As for Figure 2F, Setd2-SET activity on WT rNuc (H3) appears to be significantly lower compared to what is extensively reported in the literature. This is particularly puzzling given that Figure 2B suggests that using 3H-SAM, H3-nuc are much better substrates than K36me1, whereas in Figure 3F, rH3 is weaker than K36me1. It is recommended for the authors to perform additional experimental repeats and include a quantitative analysis to ensure the consistency and reliability of these findings.

We appreciate the Reviewer’s points. We respectfully suggest that these comments may reflect potential confusion around interpreting how different assays detect in vitro methylation, what data can and cannot be compared, and the nature of the different substrates used.

With respect to point 1 (Western signal significantly lower compared to extensive literature): To the best of our knowledge, it would be extremely challenging to make a quantitative argument comparing the strength of the Western signal in Figure 2F with results reported in the literature. Specifically, comparing our results with previous studies would require (1) all the studies to have used the exact same antibodies as antibody signal intensities vary depending on the specific activity and selectively of a particular antibody and even its lot number, (2) similar in vitro methylation reaction condition, (3) the same type of recombinant nucleosomes used, and so on. Further, given that these are Western blots, we do not understand how one could interpret an absolute activity level. In the figure, all we can conclude is that in in vitro methylation reactions, our recombinant SETD2 protein methylates rNucs to generate mono-, di-, and tri-methylation at K36 (using vetted antibodies (see Fig. 2e)). If there is a specific paper within the extensive literature that the Reviewer highlights, we could look more into the details of why the signals are different (our guess is that any difference would largely be due to the use of different antibodies). We add that it might be challenging to find a similar experiment performed in the literature; we are not aware of a similar experiment.

With respect to comparing Figure 2B and 2F: We do not understand how one can meaningfully compare incorporation of radiolabeled SAM to antibody-based detection on film using an antibody against specific methyl states. In particular, regarding the question regarding comparing rH3 vs H3K36me1 nucleosomes, we point out that in using recombinant nucleosomes installed with native modifications (e.g. H3K36me1), in which the entire population of the starting material is mono-methylated, then naturally the Western signal with an anti-H3K36me1 antibody will be strong. In Fig. 2b, the assay is incorporation of radiolabeled methyl, which is added to the preexiting mono-methylated substrate. In other words, the results are entirely consistent if one understands how the methylation reactions were performed, how methylation was detected, and the nature of the reagents.

(2) The additional bands observed in Figure 4B, which appear to be H4, should be accompanied by quantification of the intensity of the H3 bands to better assess K36me3 activity. Additionally, the quantification presented in Figure 4C for SAH does not seem accurate as it potentially includes non-specific methylation activity, likely from H4. This needs to be addressed for clarity and accuracy.

We thank the reviewer for this comment. The additional bands observed in Figure 4B represent degradation products of histone H3, not H4 methylation. This is commonly seen in in vitro reactions using recombinant nucleosomes, where partial proteolysis of H3 can occur under the assay conditions.

(3) In Figure 4E, the differences between bound and unbound substrates are not sufficiently pronounced. Given the modest differences observed, authors might want to consider repeating the assay with sufficient replicates to ensure the results are statistically robust.

In Figure 4E, we observe a clear difference between the bound and unbound substrate. To aid interpretation, we have clarified in the figure where the bound complex migrates on the gel, while the unbound nucleosomes migrate at the bottom of the gel. The differences are indeed subtle, which we highlight in the text.

(4) Regarding labeling, there are multiple issues that need correction: In the depiction of Epicypher's dNuc, it is crucial to clearly mark H2B as the upper band, rather than ambiguously labeling H2A/H2B together when two distinct bands are evident. In Figure 3B and D, the histones appear to be mislabeled, and the band corresponding to H4 has been cut off. It would be beneficial to refer to Figure 3E for correct labeling to maintain consistency and accuracy across figures.

Thank you for pointing this out. To avoid any confusion, we have delineated the H2B and H2A markers and indicate the band corresponding to H4.

(5) There are issues with the image quality in some blots; for instance, Figure 2EF and Figure 2D exhibit excessive contrast and pixelation, respectively. These issues could potentially obscure or misrepresent the data, and thus, adjustments in image processing are recommended to provide clearer, more accurate representations.

Contrast adjustments were applied uniformly across each entire image and were not used to modify any specific region of the blot. We have corrected the issue of increased pixelation in Figure 2D.

(6) The authors are recommended to provide detailed descriptions of the materials used, including catalog numbers and specific products, to allow for reproducibility and verification of experimental conditions.

We have added the missing product specifications and catalog numbers to ensure clarity and reproducibility of the experiments.

(7) The identification of Setd2 as a tumor suppressor in KrasG12C-driven LUAD is a significant finding. However, the discussion on how this discovery could inspire future therapeutic approaches needs to be more balanced. The current discussion (Page 10) around the potential use of inhibitors is somewhat confusing and could benefit from a clearer explanation of how Setd2's role could be targeted therapeutically. It would be beneficial for the authors to explore both current and potential future strategies in a more structured manner, perhaps by delineating between direct inhibitors, pathway modulators, and other therapeutic modalities.

SETD2 is a tumor suppressor in lung cancer (as we show here and many others have clearly established in the literature) and thus we would recommend avoiding a SETD2 inhibitor to treat solid tumors, as it could have a very much unwanted affect. Our discussion addresses a different point regarding the relative importance of the enzymatic activity versus other, nonenzymatic functions of SETD2. We believe that a detailed exploration of the therapeutic potential of inhibiting SETD2 would be better suited in a review or a more therapy-focused manuscript.